# Clinical Significance of COVID-19 and Diabetes: In the Pandemic Situation of SARS-CoV-2 Variants including Omicron (B.1.1.529)

**DOI:** 10.3390/biology11030400

**Published:** 2022-03-04

**Authors:** Akiko Yonekawa, Nobuyuki Shimono

**Affiliations:** 1Department of Medicine and Biosystemic Science, Graduate School of Medical Sciences, Kyushu University, Fukuoka 812-8582, Japan; 2Department of General Internal Medicine, Kyushu University Hospital, Fukuoka 812-8582, Japan; shimono.nobuyuki.679@m.kyushu-u.ac.jp; 3Center for the Study of Global Infection, Kyushu University Hospital, Fukuoka 812-8582, Japan

**Keywords:** SARS-CoV-2, COVID-19, diabetes, hyperinflammation, hyperglycemia, new-onset diabetes, Omicron variant

## Abstract

**Simple Summary:**

Amidst the dual pandemics of diabetes and coronavirus disease 2019 (COVID-19), with the constant emergence of novel variants of severe acute respiratory syndrome coronavirus 2 (SARS-CoV-2), a vicious cycle has been created, i.e., a hyperglycemic state contributes to the severe clinical course of COVID-19, which in turn has deleterious effects on glycometabolism and in some cases causes new-onset diabetes. Here, we present a comprehensive review of the current literature on the clinical and experimental findings associated with the interrelationship between diabetes and COVID-19. To control disease outcomes and glucometabolic complications in COVID-19, this issue is still being investigated.

**Abstract:**

The coronavirus disease 2019 (COVID-19) global pandemic, which is caused by severe acute respiratory syndrome coronavirus 2 (SARS-CoV-2), remains uncontrolled, with the spread of emerging variants. According to accumulating evidence, diabetes is one of the leading risk factors for a severe COVID-19 clinical course, depending on the glycemic state before admission and during COVID-19 hospitalization. Multiple factors are thought to be responsible, including an altered immune response, coexisting comorbidity, and disruption of the renin-angiotensin system through the virus–host interaction. However, the precise underlying mechanisms remain under investigation. Alternatively, the focus is currently on the diabetogenic and ketosis-prone potential of SARS-CoV-2 itself, even for probable triggers of stress and steroid-induced hyperglycemia in COVID-19. In this article, we present a comprehensive review of the recent literature on the clinical and experimental findings associated with diabetes and COVID-19, and we discuss their bidirectional relationship, i.e., the risk for an adverse prognosis and the deleterious effects on glycometabolism. Accurate assessments of the incidence of new-onset diabetes induced by COVID-19 and its pathogenicity are still unknown, especially in the context of the circulation of SARS-CoV-2 variants, such as Omicron (B.1.1.529), which is a major challenge for the future.

## 1. Introduction

Among the global outbreaks of the modern age, coronavirus disease 2019 (COVID-19) has become second only to the Spanish flu in terms of the number of deaths due to respiratory virus diseases [1]. Until 14 February 2022, i.e., in a little more than 2 years after the outbreak was announced in Wuhan, there have been 410,565,868 confirmed cases of COVID-19 worldwide, including 5,810,880 deaths (WHO Coronavirus (COVID-19) Dashboard. Available online: https://covid19.who.int (accessed on 14 February 2022). Despite global efforts for disease control, COVID-19 remains uncontrolled, and the constant emergence of novel variants of severe acute respiratory syndrome coronavirus 2 (SARS-CoV-2) make the problem even more intractable. Individuals infected with SARS-CoV-2 present with a broad spectrum of clinical manifestations, from asymptomatic to lethal. This suggests that not only the virus antigenicity but also host factors, such as age, sex, underlying health conditions, and genetic characteristics, may affect the natural history of COVID-19.

Diabetes is highly prevalent. In 2019, the global prevalence of diabetes in adult people aged 20–79 was estimated to be 9.3% (463 million people), which is predicted to continue increasing [2]. Diabetes is a heterogeneous disease that is divided into several subtypes as follows: type 1 diabetes (due to autoimmune β-cell destruction), type 2 diabetes (due to a progressive loss of adequate β-cell insulin secretion frequently on the background of insulin resistance), specific types due to other causes, and gestational diabetes [3]. The association between diabetes and infection is classically well known, and diabetes is an established risk factor for contracting infectious diseases with a high frequency and increased severity [4,5].

Diabetes has been shown to be one of the leading risk factors for poor outcomes in COVID-19 [6,7,8,9]. Additionally, emerging data reveal that COVID-19 itself dysregulates glucose homeostasis and may develop diabetes [10]. In this review, we focus on the clinical and experimental findings associated with diabetes and COVID-19 and discuss the bidirectional interrelationships between these diseases.

## 2. Risk of Diabetes as the Underlying Disease of COVID-19

### 2.1. Risk of SARS-CoV-2 Infection in Diabetes

Initially, we must distinguish between disease susceptibility and prognosis to assess risk. Patients with diabetes are generally more susceptible to infections. This is thought to be due to hyperglycemia that causes immune cells to malfunction, such as the reduction of the ability of chemotaxis, phagocytosis, and bactericidal action of polynuclear neutrophils [4,11]. Affected diabetes [11] and poor glycemic control [5] are risks of acquiring certain infections and of aggravating viral infections, including SARS [12,13] and pandemic influenza (H1N1 pdm09) [14].

The high prevalence of diabetes was reported in several studies (33.8% [15], 26.6% [16], 24% [17], and 19% [18]). A large study that included more than 5,000 individuals in the New York City reported high prevalence rates of diabetes (22.6%) [19]. However, the prevalence of diabetes in non-admitted patients was 9.7% compared with 34.7% in admitted patients [19]. The Centers for Disease Control and Prevention reported that participants, including outpatients, had a prevalence of diabetes of 10.9% compared with 10.1% of diagnosed diabetes among adults in the United States based on 2018 data [20]. Similarly, an observational study in Japan showed that diabetes without complications was 14.2% of registered COVID-19 patients, whereas diabetes is present in approximately 12% of people of the same generation [21]. In a meta-analysis reported by Pugliese et al., the prevalence of diabetes among COVID-19 patients was not higher than that of the general population, in China, Europe, and the United States [22]. Therefore, it is not epidemiologically obvious whether diabetes carries a significant risk of SARS-CoV-2 infection.

### 2.2. Significance of Diabetes for Increasing the Severity and Poor Outcome in COVID-19

Accumulating evidence shows that diabetes is the risk factor for the progression to severe disease in COVID-19.

An early study in China saw that the case fatality rate was threefold higher in people with pre-existing diabetes than in those without diabetes (7.3% vs. 2.3%) [6]. In a study of the whole population of England, of the 23,698 in-hospital COVID-19–related deaths, 31.4% had type 2 diabetes, 1.5% had type 1 diabetes, and 0.3% had other types of diabetes; moreover, both type 1 and type 2 diabetes were independently associated with a significant increase in COVID-19–related mortality (adjusted odds 2.86 (2.58–3.18) for type 1 diabetes and 1.80 (1.75–1.86) for type 2 diabetes) [7]. Regarding the fatal and critical care unit–treated COVID-19, a whole population study in Scotland found that the adjusted odds ratios (ORs) for diabetes were 1.395 (1.304–1.494), with 2.396 (1.815–3.163) for type 1 diabetes and 1.369 (1.276–1.468) for type 2 [8]. In a single-center prospective cohort study conducted in the United States, Gregory et al. initially reported that people with type 1 and type 2 diabetes showed similar risks for hospitalization and greater illness severity [9]. Their subsequent analyses from the data that included a higher number of cases showed that patients with type 1 diabetes had a higher risk for hospitalization than those with type 2 diabetes (adjusted ORs for type 1: 4.60, 95% confidence interval (CI) 3.04–6.98; type 2: 3.42, 95% CI 2.94–3.99) [23]. More recently, these authors assessed the risk of type 1 diabetes among pediatric patients. A total of 22 cases of pediatric patients with type 1 diabetes were included, and they had unadjusted ORs of 20.49 (95% CI 8.63–48.62) for hospitalization and 19.39 (95% CI 8.47–44.42) for greater severity, as compared with pediatric patients without diabetes [9].

### 2.3. Comorbidity of Diabetes

Diabetes often coexists with various comorbidities, such as hypertension, obesity, cardiovascular disease, and chronic kidney disease. It is highly possible that the overlapping of comorbidities has some additional effects on the risk of adverse outcomes. Particularly, there is a close relationship between obesity and type 2 diabetes, which is described as “diabesity”.

Obesity is independently associated with risk factors for severe COVID-19. In a large retrospective analysis of body mass index (BMI) in New York City, obesity, defined as BMI ≥ 30 kg/m^2^, was found to be significantly associated with increased admission to the hospital and critical care [24]. A single-center retrospective cohort study in France found that severe obesity, defined as BMI ≥ 35 kg/m^2^, was associated with an increased requirement for mechanical ventilation in the critical setting [25].

The risk of obesity has been assessed in diabetes patients. A population-based cohort study in England reported higher mortality rates in people with a BMI > 40.0 kg/m^2^ compared with individuals with a BMI of 25.0–29.9 kg/m^2^ (hazard ratio (HR) 2.33 and 1.60 for type 1 and type 2 diabetes, respectively) [26]. A U-shaped association was detected between BMI and COVID-19–related mortality (BMI of <20.0 kg/m^2^, HR 2.45 and 2.33 for type 1 and type 2 diabetes, respectively) [26]. In a nationwide multicenter observational study conducted in France (the CORONADO (Coronavirus SARS-CoV-2 and Diabetes Outcomes) study), BMI was also independently associated with the primary outcome of tracheal intubation for mechanical ventilation and/or death within 7 days of admission (OR 1.28, 1.10–1.47) in people with diabetes. When considering death on day 7, this association with BMI was not statistically significant. Of note, BMI ≥ 40 kg/m^2^ has less impact on the primary outcome than BMI 25–39.9 kg/m^2^ [27]. In an additional analysis of type 2 diabetes in the CORONADO study, the impact of obesity on COVID-19 prognosis was no longer observed among older patients aged ≥75 years [28].

The mechanisms by which obesity affects unfavorable outcomes remain obscure. The characteristic factors of obesity, such as a chronic low-grade inflammation state and an altered immune response [29], are potentially associated. Impaired respiratory function is one explanation for the increased need for mechanical ventilation support, as it arises from pulmonary restriction, decreased lung volume, and impaired lung perfusion [30]. To determine the BMI cutoff value at which patients are at risk for aggravation, as well as the pathogenicity of obesity in older adults, further studies are needed.

## 3. Role of Glycemic Control in Patients with COVID-19

### 3.1. Glycemic Control before Hospital Admission

An early study in China reported that poorly controlled diabetes with HbA1c ≥ 7.0% (53 mmol/mol) was associated with increased severity of lung lesions and a higher rate of poor clinical prognosis, including deaths, using mechanical ventilation, and admission to the intensive care unit (ICU), as compared with non-diabetes [31]. Subsequently, several studies have analyzed the association of HbA1c levels as an indicator of long-term glycemic control (Appendix A). Based on an analysis using a primary care electronic health records analytic platform in England, diabetes increased the risk of COVID-19–related death compared with nondiabetic patients (adjusted HR 1.31, 95% CI 1.01–1.26 for controlled diabetes with HbA1c < 7.5%; adjusted HR 1.95, 95% CI 1.83–2.08 for uncontrolled diabetes with HbA1c ≥ 7.5%) [32]. In another nationwide population-based cohort study in England, as the HbA1c level increased, the COVID-19–related mortality rate also increased in both type 1 diabetes and type 2 diabetes (type 1 diabetes HR 2.23 and type 2 diabetes: HR 1.61 for HbA1c of ≥10.0% compared with people with HbA1c of 6.5–7.0%) [26]. However, in the CORONADO study, Cariou et al. showed that the HbA1c level was not associated with COVID-19 severity or 7 day mortality [27]. This discrepancy might arise from the short-term prognosis. It is probable that the patient’s glycemic control status before hospital admission can affect the host immune system and diabetic complications, contributing to the risk of poor prognosis in COVID-19.

### 3.2. Hyperglycemia at the Time of Hospital Admission

Iacobellis et al. found that hyperglycemia on day 1 is the strongest predictor of chest radiographic abnormalities in hospitalized patients with COVID-19 [33]. Many studies have been reported on hyperglycemia at admission, although each study set different cutoff values for glucose levels (Appendix A).

Impaired fasting glucose levels at admission were shown to be associated with mortality [34,35,36], severity, including ICU admission [37,38,39,40,41,42], and in-hospital complications [34,43]. Klonoff et al. reported that in ICU inpatients, severe hyperglycemia (blood glucose (BG) > 13.88 mmol/L (250 mg/dL)) on admission was associated with increased mortality (adjusted HR 3.14; 95% CI 1.44–6.88) compared with BG < 7.77 mmol/L (140 mg/dL) [44]. In a retrospective study conducted in China, hyperglycemia (≥6.1 mmol/L (110 mg/dL)) upon admission was an independent risk factor for progression to critical or lethal cases among noncritical cases (HR 1.30, 95% CI 1.03–1.63, *p* = 0.026) and for in-hospital mortality in critical cases (HR 1.84, 95% CI 1.14–2.98, *p* = 0.013) [38].

Among patients with diabetes, poorly controlled hyperglycemia (BG > 11 mmol/L) may be associated with death and complications in COVID-19 [43]. Consistently, in age- and sex-adjusted nonlinear models in the CORONADO study, fasting plasma glucose (FPG) at admission was positively associated with tracheal intubation for mechanical ventilation and death within 7 days of admission (*p* = 0.0001) and with death on day 7 (*p* = 0.0059) [27]. Notably, a strong association with hyperglycemia at admission was observed in subjects with no history of diagnosed diabetes compared with those with known diabetes [34,40,45,46]. In addition, Fadini et al. demonstrated that a higher FPG level at admission with each 2 mmol/L (36 mg/dL) increase was associated with COVID-19 severity, with a stronger association among patients without diabetes than with diabetes (relative risk 1.21; 95% CI 1.11–1.32; *p* < 0.001) [40].

Hyperglycemia at admission, which reflects acute hyperglycemia, is a predictor of worse outcomes, even without a history of diabetes.

### 3.3. Glycemic Control during In-Hospital Treatment

Klonoff et al. reported that severe hyperglycemia (BG > 13.88 mmol/L (250 mg/dL)) on days 2–3 was an independent risk factor for high mortality (adjusted HR 7.17; 95% CI 2.62–19.62) compared with BG < 7.77 mmol/L (140 mg/dL) in non-ICU patients [44]. Wu et al. showed that a median glucose level of 6.1 mmol/L (110 mg/dL) or higher during the hospital stay or after critical diagnosis was independently associated with an increased risk for in-hospital mortality and progression to critical or lethal cases in critical cases [38]. In a multicenter retrospective observational study conducted in the United States, a subset analysis showed that the mortality rate was higher in people with uncontrolled hyperglycemia without diagnosed diabetes (>10.0 mmol (180 mg/dL) and HbA1c < 6.5%) than in those with controlled diabetes (41.7% vs. 14.8%, *p* < 0.01) [47]. In patients with type 2 diabetes, a propensity score-matched analysis in a multicenter retrospective cohort study in China demonstrated that patients with well-controlled BG (3.9–10.0 mmol/L (70–180 mg/dL)) had a significantly lower all-cause mortality rate than those with poorly controlled BG (>10.0 mmol/L (180 mg/dL); adjusted HR 0.14; CI 0.03–0.06). The well-controlled group showed lower levels of poor prognostic indicators, such as serum C-reactive protein and D-dimer, and a reduced frequency of serious complications associated with COVID-19, such as acute respiratory distress syndrome, acute heart injury, acute kidney injury, septic shock, and disseminated intravascular coagulation [48] (Appendix A).

Glycemic control has been shown to be important for improving disease prognosis. Sardu et al. demonstrated that glycemic control with insulin infusion decreased the levels of interleukin (IL)–6 and D-dimer and improved the prognosis in hyperglycemic inpatients with moderate COVID-19 [49]. They also reported that decreased BG levels from baseline to 24 h after admission were associated with a lower rate of progression to severe disease and 20 day mortality in both non-diabetic and diabetic hyperglycemic patients [50].

Poorly controlled hyperglycemia significantly increases the severity and mortality of patients with COVID-19, even after adjusting for multiple confounders (Appendix A). The treating physician should assess the patient’s glycemic status and achieve good glycemic control in patients, irrespective of pre-existing diabetes.

## 4. Factors Associated with COVID-19–Related Hyperglycemia

Worsening glycemic control during infections is common, especially in sepsis. Hyperglycemia induced by inflammation is a type of biological defense reaction against the acute stress of infectious diseases. It is known as stress hyperglycemia, which is usually defined as a transient hyperglycemic status associated with acute illness [51]. Stress hyperglycemia arises from a highly complex interplay between perturbed proinflammatory cytokines and insulin counter-regulatory hormones, which leads to the hyperproduction of hepatic glucose and the induction of insulin resistance. In addition to inflammation-induced insulin resistance, Sestan and colleagues demonstrated that virus-induced interferon-γ increases muscle insulin resistance [52].

Moreover, the use of corticosteroids for treating patients with COVID-19 strongly contributes to hyperglycemia. Following the results of the RECOVERY clinical trial [53], systemic corticosteroids have become one of the main treatment choices for COVID-19; meanwhile, systemic corticosteroids are well known to induce hyperglycemia. The main mechanism for steroid-induced glucose dysregulation is thought to be the alteration of pancreatic β-cell function, promotion of glycogenolysis in the liver, and reduction of insulin sensitivity in the liver, skeletal muscle, and adipose tissue [54]. Additionally, reduced glucose uptake and increased glycogenolysis in skeletal muscle contribute to hyperglycemia [54]. Indeed, in a multicenter retrospective study of critically ill patients with COVID-19, researchers showed that the risk of hyperglycemia (>7.8 mmol/L (>140 mg/dL)) was significantly increased with the use of steroids (OR 1.521; 95% CI 1.054–2.194) [55].

Even considering the hyperglycemia that occurs in response to acute stress or the use of glucocorticoids, the links between COVID-19 and hyperglycemia have been strongly suggested. The underlying pathogenesis of COVID-19–related hyperglycemia can be described by the damage to pancreatic β-cells and decreased insulin sensitivity and altered insulin secretion due to the activation of the renin-angiotensin system (RAS) and the host’s inflammatory response to COVID-19. The details are described below.

## 5. Effect of COVID-19 on Glucose Metabolism

### 5.1. Impaired Glucose Metabolism in COVID-19

Angiotensin-converting enzyme II (ACE2) receptors are expressed in various human organs, including pancreatic β-cells, adipose tissue, the small intestine, and the kidneys [12,56]. SARS-CoV-2 can infect pancreatic tissue and cause acute pancreatic injury [57], which results in the dysregulation of glucose metabolism. In a human pluripotent stem-cell-derived human cell and organoid model, Yang et al. found that SARS-CoV-2 infects pancreatic glucagon-positive α-cells, insulin-positive β-cells, hepatocytes, and cholangiocyte organoids [58]. Subsequently, Müller et al. showed that SARS-CoV-2 infects and replicates within the human islets of Langerhans, leading to impaired glucose-stimulated insulin secretion [59]. In addition to direct virus involvement, an enhanced autoimmune process might be implicated. Moreover, ACE2 plays an important role in regulating glucose homeostasis [60]. Downregulation of ACE2 by the entry of the virus can contribute to insulin resistance and insufficient insulin secretion through the RAS. The proinflammatory cytokines induced by COVID-19 also increase insulin resistance. Finally, hyperglycemia itself exacerbates β-cell dysfunction, leading to further poor glycemic control status.

### 5.2. New-Onset Diabetes

The development of virus-related diabetes is rare among natural viral infections; however, viral infections have long been suggested as potential environmental factors that trigger diabetes [3,61]. Particular viruses, such as enterovirus, especially coxsackievirus B virus, rubella virus, cytomegalovirus, mumps [62], and SARS-CoV-1 [12], have been associated with β-cell destruction and the triggering of type 1 diabetes; moreover, the hepatitis C virus has been implicated in inducing type 2 diabetes [63]. The putative pathogenic mechanisms of virus-related diabetes include direct β-cell destruction, local inflammatory responses, the triggering of autoimmunity against β-cells [62], and host genetic factors [64,65]. These potential mechanisms are yet to be fully studied.

There are case reports describing diabetic ketoacidosis (DKA) in a previously healthy man with no evidence of insulin resistance [66] and insulin-dependent diabetes presenting with DKA in the absence of autoantibodies 5–7 weeks after the COVID-19 episode [67]. A case series of hyper-glycemic emergencies from the United Kingdom found 2 cases with new presentation of diabetes among 35 COVID-19 patients; moreover, nine cases had type 2 diabetes among 11 patients presenting with DKA [68]. Two retrospective cohort studies in China identified patients with new-onset diabetes. In these studies, new-onset diabetes was defined as no prior history of diabetes with FPG ≥ 7.0 mmol/L [45] or random BG > 11.1 mmol/L [69] and HbA1c < 6.5%; however, the type of diabetes was not reported. The prevalence of new-onset diabetes was 5.5% [45] and 27.5% [69] in 25 of 453 and 22 of 80 hospitalized COVID-19 patients, respectively.

A recent big data analysis based on medical claim databases in the United States observed that the individuals aged <18 years with COVID-19 were more likely to be newly diagnosed with diabetes in >30 days after infection than those without COVID-19 and those with non–SARS-CoV-2 respiratory infections [70]. Some multicenter studies found an increase in the incidence of pediatric type 1 diabetes and the frequency of severe DKA [71,72,73]. However, epidemiological evidence from European studies showed no clear increase in the incidence of type 1 diabetes during the COVID-19 pandemic [74,75,76,77,78]. Moreover, Hippich et al. revealed no association between SARS-CoV-2 antibodies and autoimmunity with type 1 diabetes [79].

It is unclear whether the phenotype of the new-onset diabetes triggered by COVID-19 is the classic type 1, or type 2, or a new type and whether its alterations of glucose metabolism are transient or persistent [10]. A global registry of patients with COVID-19–related new-onset diabetes has been established (CoviDIAB Project, covidiab.e-dendrite.com) to investigate the epidemiological features and pathogenesis [10].

The expression of ACE2 in human pancreatic β cells is controversial [58,80,81], and the direct cell dysfunction cannot fully explain a key pathogenic mechanism of SARS-CoV-2 infection-induced diabetes. In COVID-19 patients, insulin resistance seems to be abnormal when compared to patients with other critical conditions [82]. This could be a factor affecting the process of new-onset hyperglycemia or diabetes in COVID-19. Montefusco and colleagues demonstrated hyperglycemia and insulin resistance in COVID-19 patients without diabetes [83]. The persistence of hyperglycemia [83] and insulin resistance [84] were documented six months after acute COVID-19 infections. Furthermore, regarding glucose homeostasis in patients with coronavirus infection, the gut is less focused than pancreas, liver, skeletal muscle, and adipose tissue. However, increased intestinal glucose absorption via the sodium-dependent glucose transporter (SGLT1) in the intestinal epithelium, which is mediated by the downregulation of ACE2 with a SARS-CoV-2 infection, might also be involved in hyperglycemia in COVID-19 patients [85].

Assessing the diabetogenic and ketosis-prone potential of SARS-CoV-2 variants is an additional challenge. The pandemic variants of SARS-CoV-2 have been reported to change the viral characteristics, including the transmissibility and antigenicity (summarized in Table 1). However, most of the currently available data were based on studies conducted before the major variants of SARS-CoV-2 were circulated. The Omicron variant is associated with a more attenuated disease severity compared to previous circulating variants [86,87]. Its cellular entry is reported to be less dependent on TMPRSS2 [88]. Replication is reduced in the lower respiratory tract that highly expresses TMPRSS2, whereas the infectivity of the Omicron variant is not affected in non-TMPRSS2-expressing cells in the upper respiratory tract [88]. Moreover, the Omicron variant is characterized by a higher ACE2 binding affinity and increased immune evasion [88,89]. The expressions of ACE2 and TMPRSS2 under steady state conditions have different distributions [58,80,90]. The tissue tropism of variants and their impact on glycometabolism might be affected.

## 6. Mechanism of Disease Aggravation in Diabetes

### 6.1. Altered Host Immune Response in the Diabetic Host

In people with diabetes, dysfunctional immune responses appear to contribute to disease progression in COVID-19. In these individuals, innate immunity is impaired, including the capacity for chemotaxis, migratory response, and phagocytosis [11]. Both type 1 and type 2 diabetes result in altered cytokine secretion and low-grade systemic inflammation status [101]. Natural killer cell activity is reduced in type 2 diabetes, in association with glucose control [102]. As for adaptive immunity, the humoral immune response is relatively normal, whereas the cell-mediated immune response is involved. Abnormal T-cell differentiation has been reported in patients with type 2 diabetes. Jagannathan-Bogdan et al. demonstrated a elevations in the Th17 subpopulation and in IL-17 and interferon-γ production, which is supported by monocytes, using human peripheral blood samples [103]. Deceased numbers of Th2 and Treg cells are reported to feature in diabetes [104]. In addition to the evidence provided by the analysis of peripheral blood mononuclear cells, Tang et al. demonstrated in a non-obese diabetic mouse model that CD4+ Foxp3+ regulatory T cells were decreased in inflamed islets during the progression of type 1 diabetes [105]. Taken together, T cells are naturally skewed toward the proinflammatory phenotype in individuals with diabetes. This status potentially exacerbates hyperinflammation, resulting in a loss of homeostasis in the inflammatory response.

In the context of COVID-19, after the recognition of SARS-CoV2 entry, proinflammatory cytokines and chemokines are generated. These attract monocytes, macrophages, and T cells to the infection site, which leads to further inflammation. In a healthy immune response, the initial inflammation, followed by the virus-specific T-cell attraction and production of neutralizing antibodies, leads to clearance of the virus before viral spreading and minimal lung damage. However, in a defective immune response, the accumulation of immune cells in the lungs, which causes an overproduction of proinflammatory cytokines, results in lung and multiorgan damage [106].

### 6.2. Role of the Renin-Angiotensin Aldosterone System

Zhou et al. [107] and Hoffmann et al. [108] identified ACE2 as a functional receptor of SARS-CoV-2. This is the same cell receptor for SARS- CoV-1 [109]. Moreover, Hoffmann et al. showed that receptor-mediated virus entry depends on cellular serine protease transmembrane serine protease 2 (TMPRSS2), which primes the spike protein. ACE2 is a membrane protein that cleaves angiotensin 2 (Ang-II) and generates angiotensin (1–7), exerting a negative effect on Ang-II signaling. Ang-II acts not only toward vasoconstriction but also toward proinflammation via the angiotensin type 1 receptor (AT1R) [110]. Meanwhile, Ang (1–7) acts toward anti-inflammation by binding to and activating the Mas receptor. After the virus binds to ACE2, ACE2 is endocytosed together with the virus complex, which reduces ACE2 expression on the cell surface, resulting in Ang-II accumulation and a decreased effect of the ACE2/Ang-(1–7)/MasR axis. This competing effect of two axes could lead to acute lung injury, cytokine storm, and ARDS [111].

In a non-obese diabetic mouse model, ACE was found to be highly expressed in the lungs, whereas ACE2 was highly expressed in the pancreas [112]. A microarray analysis revealed the upregulation of ACE2 in human pancreatic islets from diabetic donors [113]. From these results, the change of these expression levels might contribute to the increasing lung damage and β-cell tropism in diabetic patients with COVID-19, although the clinical implications remain to be established.

### 6.3. Other Host Factors

Other possible factors that increase the severity of COVID-19 in diabetes include endothelial dysfunction [114,115], a hypercoagulative state [115,116], and alveolar dysfunction [117]. Hyperglycemia enhances coagulation during systemic inflammation [118]. Varga et al. found direct endothelial cell infection by SARS-CoV-2 and subsequent endothelitis, resulting in a hypercoagulable state [119]. This might be linked to the increase in the incidence of thrombotic events during COVID-19. Alveolar dysfunction may contribute to a very low tolerance for lung damage, leading to an increased oxygen requirement.

### 6.4. Factors Affecting Viral Replication

Codo et al. demonstrated one of the mechanisms by which people with uncontrolled hyperglycemia develop a severe form of COVID-19. In human monocytes, elevated glucose levels can directly promote SARS-CoV-2 replication, the production of proinflammatory cytokines, and subsequent T-cell dysfunction and lung epithelial cell death via the production of mitochondrial reactive oxygen species and hypoxia-inducible factor-1α (HIF-1α) stabilization. SARS-CoV-2 promotes the transition to aerobic glycolysis in monocyte metabolism, which sustains the SARS-CoV-2–induced monocyte response and viral replication [120]. However, Wing et al. reported the negative effect of HIF signaling on SARS-CoV-2. In lung epithelial cells, HIF activation reduces the expression of ACE2 and TMPRSS2 and inhibits SARS-CoV-2 entry. Additionally, it inhibits viral replication and the secretion of infectious particles in an HIF-1α–dependent manner [121]. Further studies are needed to determine the role of the HIF-1α pathway and in vivo behavior.

## 7. Treatment of COVID-19 in Diabetes

Current treatment options for COVID-19 are divided into antiviral agents (e.g., remdesivir), monoclonal antibodies (e.g., casirivimab plus imdevimab and sotrovimab), immunomodulators (e.g., dexamethasone, baricitinib, and tosirizumab), and anticoagulant therapy. Recently, molnupiravir was approved as the first oral antiviral medicine for adult patients with mild-to-moderate COVID-19 and at least one risk factor associated with poor disease outcomes. Although there is no definitive therapy, these drugs are sometimes used in combination according to disease severity, oxygen requirements, and hospitalization in clinical practice. Basically, antiviral agents and monoclonal antibodies are administered against viral replication in the early stage, and anti-inflammatory drugs are administered against a dysregulated host immune response in the later stage [122].

Although there is no specific treatment for COVID-19 in patients with diabetes, prompt intervention is required because these patients are at a high risk of COVID-19 clinical progression. In Japan, monoclonal antibodies are recommended for administration as soon as possible after diagnosis to reduce the risk of hospitalization and death in patients with mild-to-moderate COVID-19 who have certain risk factors for disease progression. These risk factors include age ≥65 years, diabetes, obesity (BMI > 30 kg/m^2^), cardiovascular disease or hypertension, and chronic lung diseases (National Institutes of Health COVID-19 Treatment Guideline. Available online: https://www.covid19treatmentguidelines.nih.gov (accessed on 7 December 2021)). Further studies are needed to determine whether the use of monoclonal antibodies can improve the prognosis in diabetes. The reduction in the neutralization efficacy of imdevimab and casirivimab against the Omicron variant has been reported [89], and it is necessary to obtain up-to-date information about the epidemic variant before initiating monoclonal antibody treatment. Dexamethasone has a potential anti-inflammatory action that can prevent or suppress these excessive inflammatory responses in COVID-19, and it has been reported to have benefits for patients who received supplemental oxygen [53]. Dexamethasone should be used with more attention to glycemic control, irrespective of the diabetes status [123]. Oral Janus kinase inhibitors or anti-IL-6 receptor monoclonal antibodies might be effective for improving clinical outcomes in more severe cases. Such anti-inflammatory therapy can have negative effects on viral elimination in the host defense; therefore, it is important to assess the state of COVID-19 patients and optimize the timing of administration and drug type in high-risk patients.

Although there is no consensus on the use or discontinuation of certain antihyperglycemic agents, patients should maintain good glycemic control. A few anti-diabetic agents have been shown to influence ACE2 expressions in animal models [112,124,125,126]. Moreover, several agents not only have glucose-lowering effects but also have pleiotropic anti-inflammatory actions [127,128,129,130,131], which attract attention due to their potential beneficial roles in SARS-CoV-2 infections. However, no conclusive data demonstrate clinically significant differences (Table 2). Great caution is required when considering the possible risks resulting from lactic acidosis due to metformin as well as euglycemic diabetic ketoacidosis and dehydration due to sodium-glucose-co-transporter 2 inhibitors.

Hence, it is extremely important for people with diabetes to take steps to prevent infection, such as using hand hygiene, wearing a mask, and avoiding contact with COVID-19 patients. Vaccination against SARS-CoV-2 is recommended and can be effective in people with diabetes, as supported by the data showing no difference in humoral immune response against SARS-CoV-2 in patients with or without diabetes [132,133].

**Table 2 biology-11-00400-t002:** Potential roles of antidiabetic agents and related clinical data in COVID-19 patients.

Antidiabetic Agent	Effects on ACE2 Expression in Animal Models	Anti-Inflammatory Properties	Data from Clinical Studies on COVID-19	Clinical Considerations
**Metformin**	–	+ [127]	No association [134,135]	Risk of lactic acidosis Discontinued in severe cases
			Lower incidence of COVID-19 and no effects on mortality [136]	
			Improved outcomes [137]	
			Reduced mortality [27,138,139], only in women [140]	
**Pioglitazone**	Upregulation of ACE2 expressions in the liver, adipose tissues, and skeletal muscles in NASH rats [124]	+ [128]	Insufficient evidence	Risk of fluid retention and heart failure Discontinued in severe cases
**SGLT2 inhibitors**	–	+ [129]	No influence on susceptibility compared with DPP-4 inhibitors [141]	Risk of euglycemic diabetic ketoacidosis and dehydration Discontinued in severe cases
**GLP1-RAs**	Upregulation of ACE2 expression in lungs in STZ-induced diabetic rats [125]	+ [129,130]	Insufficient evidence	Risk of dehydration in cases with severe gastrointestinal symptoms
**DPP-4 inhibitors**	–	+ [130,131]	Not associated with adverse outcomes [134,142,143,144]	Relatively safe
			Lower mortality [145,146]	
			Improved outcomes [146]	
			Worse outcome [147]	
**Insulin**	Normalization of renal ADAM17 and ACE2 expressions in diabetic Akita mice [126]	+; reduced inflammatory marker [148]	Poor prognosis [149]	Risk of hypoglycemia Require frequent blood glucose monitoring
			Increased mortality [27,137,145,150]	
	Increased ACE2/ACE in NOD mice [112]		No association [134,139]	
			Beneficial effects of control during hospitalization [49]	

SGLT2 inhibitors, Sodium-glucose cotransporter-2 inhibitors; GLP1-RAs, Glucagon-like peptide 1 receptor agonists; DPP-4 inhibitors, Dipeptidyl peptidase-4 in-hibitors; ACE2, angiotensin-converting enzyme 2; ACE, angiotensin-converting enzyme; NASH, Nonalcoholic Steatohepatitis; STZ, Streptozocin; NOD, Non-obese diabetic.

## 8. Conclusions

We are amidst dual pandemics of COVID-19 and diabetes and face the threat of the emergence of SARS-CoV-2 variants, including the Omicron variant. These are intricately interrelated, and it is particularly important to reveal the precise mechanism underlying the association between COVID-19 and diabetes as well as to dissolve the vicious circle of hyperinflammation and hyperglycemia. The incidence and etiology of new-onset diabetes remain to be fully elucidated, and the diabetogenic properties of SARS-CoV-2 depending on the variants is a subject for future investigation.

## Figures and Tables

**Table 1 biology-11-00400-t001:** Comparison of the characteristics of pandemic SARS-CoV-2 variants.

WHO Label	Alpha	Delta	Omicron
Pango Lineage	B.1.1.7, Q	B.1.617.2, AY.4.2	B.1.1.529, BA
**First detected**	United Kingdom	India	South Africa
	September 2020	December 2020	November 2021
**Key amino acid substitutions in spike protein**	N501Y, D614G, and P681H	L452R, T478K, D614G, and P681R	30 changes, 3 small deletions, and 1 small insertion *
**Infectivity Transmissibility**	↑	↑~↑↑	↑↑↑
	43–82% more transmissible than the ancestral lineage	At least equal to alpha	Rt_Omicron_ 3.19–4.2 times greater than Rt_Delta_, with higher secondary attack rates
**Clinical severity**	→/↑	↑↑	↓
	60% higher mortality than the ancestral lineage	risk for hospital admission twice as high as alpha	Decrease in severity and mortality
**Immune escape**	→	↑	↑↑
		Higher reinfection rate and reduced vaccine efficacy	
**Diabetogenecity**	Insufficient data **	No data	No data

* A67V, del69-70, T95I, del142-144, Y145D, del211, L212I, ins214EPE, G339D, S371L, S373P, S375F, K417N, N440K, G446S, S477N, T478K, E484A, Q493R, G496S, Q498R, N501Y, Y505H, T547K, D614G, H655Y, N679K, P681H, N764K, D796Y, N856K, Q954H, N969K, and L981F (RBD substitutions in bold type). ↑: increased; ↓: decreased; Rt: effective reproduction number. ** A study supporting new-onset diabetes [70] included data during the period when the alpha variant was dominant. References: [91], transmissibility [92,93,94], Rt [95,96], secondary infection rate [97], severity [86,92,98,99], reinfection [100], and immune escape [89].

## Data Availability

Not applicable.

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
