# Peer review of "Clinical Significance of COVID-19 and Diabetes: In the Pandemic Situation of SARS-CoV-2 Variants including Omicron (B.1.1.529)"

_biology, 2022, doi:10.3390/biology11030400_

Round 1
Reviewer 1 Report
Suggestions:
- Include more information about the reviewed content and key observations in the abstract.
- SARS-CoV-2 infection risk in diabetics: Don't conclude itself as it's not a systematic review. Discuss the evidence-based observations in pro and against and underline the dominant view.
- New onset of diabetes: It's a controversial topic hence needs more discussion presenting alternative views.
- Possible mechanisms of new diabetic onset: Possible role of ACE2 mediated dysregulation of intestinal glucose transporters can be discussed: https://pubmed.ncbi.nlm.nih.gov/33254575/
- Therapeutic management of COVID-19 in diabetics: This section needs to be discussed in greater details.
Reviewer 2 Report
Comments
1. Citation of references is needed throughout the entire manuscript. For example:
“The association between diabetes and infection is classically well known, and diabetes is an established risk factor for the contraction of infectious diseases with high frequency and increased severity”.
“Diabetes has been shown to be one of the leading risk factors for poor outcomes in COVID-19”.
“People with diabetes are generally more susceptible to infections, a fact that is considered to be due to hyperglycemia, which causes the malfunction of the immune cells, such as the reduction of the ability of chemotaxis, phagocytosis, and bactericidal action of polynuclear neutrophils”.
2. On which reference is table 1 based?
Round 2
Reviewer 1 Report
No issues.
Author Response
We thank you for taking the time and energy to help us improve the paper.
In accordance with the academic editor’s comment, we have added “in adult people aged 20-79” to “the global prevalence of diabetes”(section 1., p. 3, lines 70–71, highlighted in yellow).
